# HSSG: Identification of Cancer Subtypes Based on Heterogeneity Score of A Single Gene

**DOI:** 10.3390/cells11152456

**Published:** 2022-08-08

**Authors:** Shanchen Pang, Wenhao Wu, Yuanyuan Zhang, Shudong Wang, Muyuan Niu, Kuijie Zhang, Wenjing Yin

**Affiliations:** 1College of Computer Science and Technology, Qingdao Institute of Software, China University of Petroleum, Qingdao 266580, China; 2School of Information and Control Engineering, Qingdao University of Technology, Qingdao 266525, China; 3Normal College, Qingdao University, Qingdao 266071, China

**Keywords:** cancer subtypes, heterogeneity, single gene, pseudo-F statistic

## Abstract

Cancer is a highly heterogeneous disease, which leads to the fact that even the same cancer can be further classified into different subtypes according to its pathology. With the multi-omics data widely used in cancer subtypes identification, effective feature selection is essential for accurately identifying cancer subtypes. However, the feature selection in the existing cancer subtypes identification methods has the problem that the most helpful features cannot be selected from a biomolecular perspective, and the relationship between the selected features cannot be reflected. To solve this problem, we propose a method for feature selection to identify cancer subtypes based on the heterogeneity score of a single gene: HSSG. In the proposed method, the sample-similarity network of a single gene is constructed, and pseudo-F statistics calculates the heterogeneity score for cancer subtypes identification of each gene. Finally, we construct gene-gene networks using genes with higher heterogeneity scores and mine essential genes from the networks. From the seven TCGA data sets for three experiments, including cancer subtypes identification in single-omics data, the performance in feature selection of multi-omics data, and the effectiveness and stability of the selected features, HSSG achieves good performance in all. This indicates that HSSG can effectively select features for subtypes identification.

## 1. Introduction

Cancer is a complex and highly heterogeneous disease caused by many factors, and its heterogeneity makes it challenging to target specific therapies for different tumor types [1,2]. The clinical heterogeneity of cancer is traceable to the discovery that morphologically similar tumors have several subtypes with distinct pathogeneses [3,4]. Therefore, the accuracy of the cancer subtype is not only meaningful in the development of individualized treatment strategies for patients and understanding of the evolution of the cancer [5,6,7,8]. More importantly, identifying cancer subtypes can provide an understanding of the underlying molecular mechanisms that can lead to the design of precise therapeutic strategies for effective cancer management.

With the development of high-throughput sequencing technologies, a large amount of multi-omics data on various cancers has been generated. A large number of cancer subtype identification efforts based on multi-omics data have been developed [9,10,11,12,13,14]. For example, Guo et al. used a denoising autoencoder to fuse multi-omics data for ovarian cancer subtype identification [10]. Tang et al. proposed a method based on statistical moments to reduce the dimension of high-throughput protein omics data to identify cancer subtypes [12]. Wasito et al. used the combination of kernel classification and SVM to reduce the dimension of patient gene expression data and realize the subtypes identification of lymphoma [14]. However, in using multi-omics data for cancer subtype identification, the sheer number of genes always results in meaningful information being overwhelmed by redundant or noisy data. Therefore, deriving or selecting the most discriminatory subset of genes from tens of thousands of genes is necessary to identify cancer subtypes and reduce computational costs [15]. So far, most studies have focused on two types of approaches: feature extraction (FE) and feature selection (FS). The major difference between FS and FE is that FS retains the original features, whereas FE achieves dimensionality reduction by compression or mapping. In biology-related studies, exploring the biology mechanism always requires the identification of marker genes, so FS on biomolecules is more often used than FE [16]. Nowadays, there are many FS-related methods, such as the most commonly used differential expression-based gene screening, the Kruskal–Wallis Test, entropy-based methods, and random forests, among others. However, most of the existing studies do not consider fusion for each cancer sample specificity from the biomolecular perspective, but they are based only on differences in numerical patterns of multi-omics data.

Cancer samples are highly heterogeneous and even patients with the same cancer directly have their unique characteristics [17]. The heterogeneity of cancer samples plays an important role in understanding and studying the process of cancer development and its pathogenesis [18,19,20]. Based on this idea, a small amount of work has applied the specificity of cancer samples to the identification of cancer subtypes [21,22,23]. For example, Chen et al. combined gene expression data and gene interaction networks to identify breast cancer subtypes by sample-specific perturbations in the biological network [21].

Zhang et al. proposed a cancer subtype identification method based on single-sample information gain (SSIG), which fused multi-omics data by considering the heterogeneity of samples to identify breast cancer (BRCA) and kidney clear cell carcinoma (KIRC) [22]. Nakazawa et al. proposed an approach based on sample-specific molecular regulatory systems to identify cancer subtypes, with good results on lung, gastric, and breast cancer datasets [23]. The above studies indicate that the specificity of cancer samples is important in the identification of cancer subtypes. However, no studies have yet adequately considered cancer specificity in biomolecules.

A novel method is proposed for feature selection to identify cancer subtypes based on the heterogeneity score of a single gene: HSSG. The proposed method takes full account of each sample’s specificity under biomolecules, considering the specific features of cancer samples under individual genes. In addition, the linkage between the selected genes is also considered, aiming to select a better subset of genes for cancer subtype identification. Firstly, a sample similarity network under each gene is constructed based on the differential expression vector of the cancer sample and normal samples. This takes into account not only the specific characteristics of each cancer sample but also the specificity of the cancer samples effectively fused under the genes. Secondly, the pseudo-F statistic is used to calculate the heterogeneity score of a single gene for cancer subtypes identification. Finally, a gene–gene network is constructed and analyzed to find possible functional modules, and the critical genes for cancer subtypes identification are selected. Experiments on the TCGA data sets show that HSSG has achieved better performance than other popular feature/gene scoring methods. The overall framework is shown in Figure 1.

## 2. Materials and Methods

### 2.1. Data Sources

From the TCGA hub data set of UCSC Xena [24], we obtained RNA-seq data, miRNA-seq data, and the clinical data of five types of cancer samples, including lung cancer (which is divided into lung squamous cell carcinoma (LUSC) and lung adenocarcinoma (LUAD)), stomach cancer (STAD), colon cancer (COAD) and thyroid cancer (THCA). The reasons for the choice of these types of tumors: on the one hand, these types of cancer tumors chosen are very common types of cancer tumors. There has been a great deal of work done on these types of tumors. On the other hand, benefiting from the profession of the TCGA project, researchers have collected a wealth of samples that provide a solid basis for the calculations of our study. The detailed information of these data is shown in Table 1.

### 2.2. Construction of Sample-Similarity Network Based on a Single Gene

In this paper, $C_{i}^{g}$ and $S_{j}^{g}$ represent the gene expression value of *i*-th cancer sample and *j*-th normal sample under the gene *g*, respectively (i∈[1,N],j∈[1,M],N,M represent the total number of disease samples and normal samples, respectively). Firstly, we can obtain the differential expression vector $V_{i}^{g}$ of each disease sample with all normal samples in the gene *g* using the following Equation (Equation 1):(1)$$\begin{array}{c} V_{i}^{g}=C_{i}^{g}-S_{*}^{g}, \end{array}$$
where $S_{*}^{g}$ and $V_{i}^{g}$ are M×1, which respectively represents the gene expression value of all the normal samples and the differential expression between disease sample *i* under the gene *g* (i∈[1,N]). $V_{i}^{g}$ can effectively reflect the differences between cancer samples and normal samples and enrich the data characteristics of cancer samples under a specific gene.

To measure the contribution of a single gene to cancer subtype identification, the similarity of two differential expression vectors of any two cancer samples with all normal samples under gene *g* is calculated, denoted by $D^{g}$, which is a similarity network between cancer samples, using the following Equation (Equation 2):(2)$$\begin{array}{c} D^{g}={\left(d_{st}^{g}\right)}_{N\times N}=\left(|\right|V_{s}^{g}-V_{t}^{g}{\left|\right|}_{2}{)}_{N\times N}, \end{array}$$
where $V_{s}^{g}$,$V_{t}^{g}$, respectively, represent the differential expression vectors of the *s*-th and *t*-th cancer samples versus normal samples under gene *g*(s,t∈[1,N]), $\left|\right|V_{s}^{g}-V_{t}^{g}{\left|\right|}_{2}$ represents the 2-norm of the two vectors and $d_{st}^{g}$ is the similarity between *s* and *t* under gene *g*. This process for constructing the sample similarity network of a single gene is shown in Figure 2.

### 2.3. Calculating Heterogeneity Score of Single Gene Based on Pseudo-F Statistics

To describe the contribution of each gene to subtypes identification, pseudo-F statistics [25] is used to calculate the heterogeneity score of each gene for cancer subtypes identification. Specifically, pseudo-F statistics are based on subtype categories as inputs and calculated from the similarity matrix of the above-mentioned single gene similarity networks. Furthermore, it can test the correlation between genes and cancer subtype classifications. Assuming a total of *N* samples exist, the formula is as follows:(3)$$\begin{array}{c} F=\frac{tr(\mathrm{HGH})}{tr[(I-H)G(I-H)]}\;, \end{array}$$
where $H=Y{\left(Y^{T}Y\right)}^{-1}Y^{T}$ is a N×N matrix, Y is a N×1 vector that indicates *N* samples belong to which subtypes ($Y=\left(y_{m}\right)\;y_{m}\in \left[1,k\right],\;k$ indicates that there are *k* cancer subtypes), $G=\left(I-1/N11^{T}\right)A\left(I-1/N11^{T}\right)$ is the Gower’s centered matrix, $A=\left(a_{st}\right)=\left(-1/2{\left(d_{st}\right)}^{2}\right)$) ($d_{st}\in D$, D is the sample similarity matrix of a single gene). 1 is a dimensional column vector with element 1, and I is an identity matrix. The contribution of each gene for subtypes identification is calculated by using pseudo-F statistics. After calculating the contribution of all genes, we obtain a heterogeneity score for each gene. This score shows the adequate aggregation degree of different cancer subtype samples under a specific gene, which can explain the contribution of this gene to cancer subtypes identification.

### 2.4. Construction of Gene–Gene Network Based on Single Gene Sample-Similarity Network

As cancer is often affected by multiple regulatory modules, we construct a gene–gene network based on the single gene network obtained above to conform to the biological regulation law to the greatest extent. Firstly, the genes for constructing the gene-gene network are screened by the pseudo-F statistic. $\tilde{G}=\left(\tilde{g_{i}}\right)$ represents the gene order sorted according to the pseudo-F statistics value. The formula for screening the genes $\hat{G}$ that construct the network is as follows:(4)$$\begin{array}{c} \hat{G}=\left(g_{k}\right)|\left(ACC=Kmeans\left(C_{*}^{\hat{G}}\right)\right)==max\left(ACC\right), \end{array}$$
where $\hat{G}\subset \tilde{G}$, k∈[1,b] (*b* is the number of selected genes), $C_{*}^{\hat{G}}$ represents the gene expression matrix of all cancer samples under gene set $\hat{G}$, ACC refers to the accuracy of K-means clustering [26] with selected genes as features. The genes are selected with the highest clustering accuracy as the screening genes. Then, after screening genes, those genes can be used to construct the gene–gene network W, using the following formula:(5)$$\begin{array}{c} W=\left(w_{qp}\right)=\left(|\right|D^{g_{q}}-D^{g_{p}}{\left|\right|}_{F}{)}_{b\times b}, \end{array}$$
where W is b×b matrix indicates the gene–gene network, $D^{g_{q}},D^{g_{p}}$ represents sample-similar matrices under gene $g_{q}$ and gene $g_{p}$ ($g_{q},g_{p}\in \hat{G}$), respectively, and ${\left||D|\right|}_{F}$ is the F-norm of matrix D.

### 2.5. Network Analysis and Module Mining

To identify the critical functional modules of cancer subtypes, we perform the following operations on the network: firstly, we mined the nodes with the largest degrees in the network. We believe these nodes play a more critical role in the network [27,28]. Secondly, the cluster_walktrap function in R (R package igraph 1.3.0) is used to mine the functional modules in the network [29]. After getting modules by cluster_walktrap function, the modules are sorted according to the average pseudo-F statistics of each module and the ${\tilde{M}}_{i}$ is used to represents the *i*-th gene module in the sorted ordermodules (${\tilde{M}}_{i}\subset \tilde{M}$, $\tilde{M}$ represents all the sorted order modules). Then, Matthews correlation coefficient (MCC) [30] and RAND coefficient (RI) [31] are used to measure the performance of the gene modules, and MCC and RI are defined as follows:(6)$$\begin{array}{c} MCC=\frac{TP\times TN-FP\times FN}{\sqrt{\left(TP+FP)(FP+FN)(TN+FP)(TN+FN\right)}}, \end{array}$$
(7)$$\begin{array}{c} RI=\frac{TP+TN}{TP+FP+TN+FN}, \end{array}$$
where TP, FP, TN, and FN mean true positive, false positive, true negative, and false negative. Finally, we use the following formula to get the needed module genes $\hat{M}$:(8)$$\begin{array}{c} \hat{M}=\left({\hat{m}}_{i}\right)|\;if\left(\left(MCC/RI=Kmeans\left(C_{*}^{{\tilde{M}}_{i}+{\tilde{M}}_{j}}\right)\right)>\left(MCC/RI=Kmeans\left(C_{*}^{{\tilde{M}}_{j}}\right)\right)\right){\tilde{M}}_{i}\subset \hat{M} \end{array}$$
where $\hat{M}\subseteq \tilde{M}$, $C_{*}^{{\tilde{M}}_{j}}$ represents the gene expression matrix of all cancer samples under gene set ${\tilde{M}}_{j}$, MCC/RI refers to the measurement index of K-means clustering with selected genes as features. When a module is added, the corresponding gene modules for the functional module are retained if the clustering indicator increases. If the indicator is not added, the community abandons. Finally, we get the genes from module mining for the network.

### 2.6. Performance Evaluation Metrics

The performance of the proposed method is evaluated based on accuracy, sensitivity, and specificity [32]. They are defined as follows:(9)$$\begin{array}{c} \;Accuracy=\frac{TP+TN}{TP+FP+TN+FN}, \end{array}$$
(10)$$\begin{array}{c} \;\;\;Sensitivity=\frac{TP}{TP+FN}, \end{array}$$
(11)$$\begin{array}{c} \;\;\;Specificity=\frac{TN}{TN+FP}\;. \end{array}$$

At the same time, the identified gene list was imported into the GO enrichment analysis and the KEGG pathway enrichment analysis to understand the biological significance of the selected genes further. In addition, we randomly divided the cancer subtype samples into training and validation sets in a ratio of 2:1 to test the performance of HSSG. In addition, the random seeds were set to ensure the reproducibility of the experiments.

Our source code of HSSG is Available online: https://github.com/ubuntu1024/HSSG (Last visited on 1 August 2022). The main packages and R versions used in the code are: R version 4.0.5; R package limma 3.46.0; R package readr 2.1.2; R package igraph 1.3.0, etc.

## 3. Results

To verify the effectiveness of the HSSG, seven data sets of five cancer subtypes from TCGA were experimented with in three ways: Firstly, on the single omics data of RNA—seq, HSSG can effectively identify the essential genes and the subtypes in the experiments, including two subtypes identification and multiple cancer subtypes identification. Secondly, the multi-omics data before and after HSSG feature selection was applied to the five popular cancer subtypes identification methods based on the multi-omics data. Moreover, the results showed that HSSG also performed well in feature selection of multi-omics data for cancer subtypes identification. Finally, to verify the effectiveness and stability of the genes selected by HSSG, the cancer samples were divided into two halves, and the normal samples were placed into the cancer samples as subtypes. Half of the cancer samples and the normal samples were used as the training set to screen genes using HSSG, and the other half was used as the verification set to verify the effectiveness and stability of the selected genes. It can be found that the genes selected by HSSG can effectively distinguish different samples.

### 3.1. Identification of Cancer Subtypes Based on Single-Omics Data

#### 3.1.1. Result of Identifying Two Cancer Subtypes

The RNA-seq data of lung cancer, including LUAD and LUSC, was involved in this experiment (the data includes 517 samples of lung adenocarcinoma and 502 samples of lung squamous cell carcinoma). By the above method, the heterogeneity score of each gene for the identification of cancer subtypes was calculated, and the genes were sorted according to the value of pseudo-F statistics. By adding genes as features to theorderfor clustering, the curve of clustering accuracy with the number of genes is shown by the red line in Figure 3a. The red line in Figure 3a shows the clustering accuracy curve with the number of genes added, based on the ranking of each gene’s contribution (pseudo-F statistic value), with the constant addition of genes. When the number of added genes reaches 3000, the accuracy rate is the highest (95.1%). At the same time, to highlight the effectiveness of HSSG in screening essential genes that can identify cancer subtypes, we compared three different methods for screening genes. The curve of clustering accuracy of the three methods with the rank of gene addition is shown in Figure 3a. The genes selected based on the differential expression method were analyzed using the limma differential expression analysis method. The limma package in R language (R package limma 3.46.0) was used to conduct differential expression analysis on sample data of two cancer subtypes, and the obtained results were ranked according to logFC. LogFC represented the ratio of expression levels between two samples (groups). In general, the absolute value of logFC could be used as the screening criterion for differential genes. The clustering accuracy rate of random selection averaged the clustering accuracy rates of the corresponding number of genes after 50 random selections. The figure shows that the accuracy rate of gene clustering sorted by HSSG screening is higher than that of the other two methods.

Then, the first 3000 genes were used to construct a gene-gene network using the methods described above. After obtaining the gene-gene network, the weighted adjacency matrix of the network was normalized, and the threshold was set to 0.7 to remove the edges where the two genes were far apart. On the one hand, through the analysis of the largest network degrees, the largest 67 network nodes were obtained (the degree of these nodes is 2999). On the other hand, module mining also was used on the network. By setting the random walk step size to 7 in module mining, 28 gene modules could be obtained. The proposed method for rankingthe gene modules to cluster shows the precision curve in Figure 3b. And Figure 3b shows the clustering accuracy curve with module genes accessions after constructing a gene-gene network, based on network mining of functional modules and ranking modules by average module contribution. It is not difficult to see that with adding a module, the accuracy of clustering does not continuously increase, but there is continuous fluctuation after adding specific module genes. To accurately mine the key gene modules for cancer subtype identification, the Matthews correlation coefficient (MCC) was used to measure the clustering effect. Matthews correlation coefficient is one of the practical evaluation indexes to measure the result of the two-classification model. In the end, six gene modules were retained for 644 genes. The 67 genes were mined by performing a degree analysis on the constructed gene-gene network to find the gene nodes with the maximum degree. In addition, the 644 genes were mined for the more important gene modules by performing module mining of the constructed gene-gene network. Finally, through the above gene–gene network analysis and module mining, we obtained 67 key genes and 644 key genes.

To visualize the global expression changes of the selected genes in all cancer samples, the simple heat maps of the expression values in two different cancer samples of the screened 67 genes and 644 genes found by module mining were drawn, as shown in Figure 4. The heat map clearly shows that the expression values of most of the genes based on the HSSG screen are more significantly different between the LUAD and LUSC subtypes. Finally, the clinical information of the samples was used to analyze the survival rate of the clustering results. As shown in Figure 5a, it was the graph showing the change in sample survival rate after clustering the 67 genes. The *p*-value of the survival rate in the schematic diagram is 0.037 < 0.05, which indicates that there are significant differences between the two groups. As shown in Figure 5b, the graph shows the change in sample survival rate after clustering the 644 genes. The *p*-value of the survival rate in the schematic diagram is 0.029 < 0.05, indicating significant differences between the two groups. They demonstrated the superior performance of HSSG.

To find out whether the genes we found have biological significance, 67 genes and 644 genes were enriched and analyzed, and the results showed in Figure 5c,d. The result of GO enrichment analysis on 67 genes showed that these genes were mainly related to epidermis development, regulation of endopeptidase activity, negative regulation of endopeptidase in biological processes; They were related to connexin complex and gap junction in cell composition; they were related to endopeptidase inhibitor activity, peptidase inhibitor activity, endopeptidase regulator activity in molecular function; From the go enrichment analysis of the above 67 genes, these 67 genes were related to the development of epithelial cells. However, it is well known that the most significant difference between lung adenocarcinoma and lung squamous cell carcinoma is that lung squamous cell carcinoma is mainly caused by chronic irritation and injury of columnar epithelial cells of the bronchial mucosa, cilia loss, squamous metaplasia of basal cells, atypical hyperplasia and hypoplasia [33]. They showed, in some ways, the correctness of the screening genes of HSSG. Similarly, the GO enrichment analysis on 644 genes was also performed. Furthermore, the results were similar to those of the 67 genes enrichment analysis, mainly related to the development of epithelial cells.

To show the effectiveness of the proposed method, we compared its performance with the other six popular feature/gene scoring methods, which used Random selection, Differential Expression selection, Variance-based score, Kruskal–Wallis Test [34], Entropy-based score, and Random Forest as the gene importance evaluation criterion. Random selection and differential expression selection methods are consistent with the beginning of this experiment. The effects of the above methods in selecting 67 genes and 644 genes are shown in Table 2 and Table 3. It can be seen from the Table that HSSG is slightly better than other feature/gene scoring methods. To compare the performance of HSSG with other popular feature scoring methods and the degree of overlap between the features selected by the different methods, we set thresholds for the five popular methods to make them select the best features. The number of features selected, the accuracy rate, the number of overlaps between each method and the features selected by HSSG, and the overlap rate are shown in Table 4 (the Random selection method is the average number of feature overlaps taken 50 times at random). As can be seen from Table 4, HSSG can select relatively few genes while achieving the final performance. Then, the clustering heatmaps of gene expression values for the genes screened by the different methods were drawn to visually compare the effects of the different methods of gene screening, as shown in Figure 6 (the clustering heatmaps for random selection are not drawn here because the random selection method was averaged 50 times at random and the features selected at random were not consistent each time). In Figure 6, the cancer subtype samples were clustered hierarchically with genes based on the expression values of the screened genes, respectively, and the clustering results were subjected to the fisher test. As can be seen from the figure, the HSSG test with the smallest *p*-value (*p* = 2.23 × 10^−75^) obtained the best results.

In addition, 79 genes selected by Differential Expression were enriched for analysis to explore how the biological significance of the genes selected by the HSSG differs from the commonly used differential expression-based approach. The differential expression-based gene enrichment analysis is shown in Figure 7 and the HSSG screened genes enrichment analysis is shown in Figure 5. As can be seen from the figure, the results of the GO enrichment analysis based on HSSG screening genes and the results of the conventional differential expression-based GO enrichment analysis are in agreement for a large part. There are also some inconsistencies, such as embryonic organ morphogenesis in Biological Process, complex trimers in Cellular Component; gap junction in Molecular Function channel activity in Molecular Function, etc. Finally, the HSSG KEGG pathway enrichment analysis and differential expression-based KEGG pathway enrichment analysis were completed. We found that the HSSG-selected gene pathways were mainly related to Cell cycle and Cellular senescence, while the differential expression-based pathways were mainly related to the estrogen signaling pathway and *Staphylococcus aureus* infection.

#### 3.1.2. Result of Identifying Multiple Cancer Subtypes

After proving the effectiveness of HSSG in the identification of two cancer subtypes, another experiment was done to verify the recognition effect of HSSG in three or more cancer subtypes. However, due to the uncertainty of cancer subtypes, it was difficult to find definite three or more subtypes data of the same cancer. In this experiment, the samples of three different cancers (350 gastric cancer, 288 colon cancer, and 350 thyroid cancer) were mixed, and HSSG was used to verify whether different cancer samples could be effectively distinguished. As the above method, the heterogeneity score of each gene could be calculated, and the genes were sorted with the score. Then, genes were added in order as features, and K-means clustering was adopted (clustering into three classes). The change curve of clustering accuracy is shown in the red line in Figure 8. The meaning of the other two curves shown in Figure 8 is consistent with that of the above experiment. From the figure, the accuracy fluctuates first rose to the highest and then decreased with different numbers of genes added. When the number of genes was added to 500, i.e., When the top 500 genes in pseudo-F statistics were added, the accuracy reached 99.89%. To further enhance the bio-interpretability of the method and improve the gene-gene relationship, the first 500 genes were selected to construct a gene–gene network.

On the one hand, through analyzing the degree distribution of the network, the nodes with the largest degrees in the network were selected, with 90 full-degree nodes selected. And the clustering accuracy was 89.3% when the samples were clustered based on these genes as a feature. (Results from the degrees distribution analysis for the network in this experiment were not very effective, and these genes were not studied in depth later). On the other hand, module mining was performed on the network. We got six gene communities using the cluster_walktrap function and setting the step size to 7. The communities are sorted according to the average pseudo-F statistic. Then, the Rand Index (RI) was used to measure the effectiveness of clustering when joining different communities. In the end, 196 community genes could be gained with a cluster accuracy of 99.89% and RI of 0.9986. Then, the selected genes were compared with the corresponding numbers of genes from the other six different methodsselected, with the accuracy shown in Table 5. From Table 5, it is not hard to see that the effect of screening practical features of HSSG is better than the other methods.

At the same time, the simple heat maps of the expression values in three different cancer samples of the 500 genes and the 196 genes screened were shown in Figure 9. Through the heat maps, it is not difficult to see a significant difference in the expression of genes screened by HSSG in the samples of three cancer subtypes. Finally, the enrichment analysis of 196 genes is shown in Figure 10. Through enrichment analysis, it could find that most of our selected genes were related to gland development, and the KEGG pathway was also related to thyroid hormone synthesis.

### 3.2. The Performance in Feature Selection of Multi-Omics Data

To verify the effectiveness of HSSG in feature selection of multi-omics data, multi-omics data of LUAD and LUSC was used for this experiment (i.e., binding of RNA-seq data and miRNA-seq data). HSSG was used to select the features of RNA-seq data and miRNA-seq data. And the feature selection of RNA-seq data of lung cancer was completed in the above experiments. The 644 genes selected in the above experiments were used as the features of RNA-seq data. In the same way, HSSG conducted feature selection on the miRNA-seq data of lung cancer and finally selected 442 miRNAs as the features of miRNA-seq data. Then the data without HSSG feature selection and the data with HSSG feature selection were applied to five popular cancer subtypes identification methods based on multi-omics data, including COCA [35], LRAcluster [36], ConsensusClustering [37], iClusterBayes [38], and IntNMF [39], and the results are shown in Table 6. As shown in Table 6, it can be seen that the effect of each method has been improved after the feature selection of HSSG.

After that, the top 10 miRNA in the mined 442 miRNA with higher heterogeneity scores based on pseudo-F statistics were carefully studied. Through a literature search, 8 of the top 10 mined miRNAs were closely related to lung cancer, which also proved that HSSG has good performance in feature selection. The detail of the top 10 miRNA is shown in Table 7.

### 3.3. The Effectiveness and Stability of the Selected Genes

The above aspects show that HSSG could effectively select features and identify cancer subtypes. After that, the effectiveness and stability of the selected genes were verified in this experiment. In this experiment, the 400 thyroid cancer samples were divided equally into two groups of 200 samples each. Then, the first 200 cancer samples were then mixed with 59 normal samples (as a subtype) to obtain a disease set of 259 samples to identify normal and cancer samples. The heterogeneity score of each gene was calculated, and the genes were ordered according to the above method’s value. Finally, genes were added in order as features to conduct K-means clustering, and the change curve of accuracy is shown in the red line in Figure 11. At the same time, the genes screened out based on the differential expression method (limma differential expression analysis) were used as the control, and the curve of change in accuracy is shown in the blue line in Figure 11.

From Figure 11, it is not hard to find that when the number of genes was added to 40 in the order of pseudo-F statistical values, the cluster accuracy reached the maximum of **82.9%**. And the accuracy of the genes found by HSSG was higher than the genes selected based on differential expression, which further verified the method’s effectiveness. Through analysis, the reason for the low accuracy should be that the classification data of normal and cancer samples is unbalanced, leading to the low accuracy of the experiment.

Finally, as there are too fewer genes to construct gene–gene networks when the accuracy reaches the maximum, 40 genes were used as features to cluster these 259 samples, including the last 200 cancer samples and 59 normal samples. The accuracy of clustering was **81.4%**, which proved that the genes we found were still valid for other samples.

## 4. Discussion

This paper proposes a heterogeneity score of the single gene method, HSSG, to identify cancer subtypes effectively. First, the sample similarity network of a single gene is constructed based on the existing data, and the TOP gene that contributes to clustering is screened out based on the pseudo-F statistic, which improves the accuracy of the experiment. The distance between the two genes is then calculated to build a network of gene interactions under a particular disease. Through the network analysis, the key genes of subtypes recognition are found. In addition, K-means are clustered by using the key genes as the characteristics. We did experiments from three aspects, and the results showed that our method had a significant effect on the classification of cancer subtypes and mining the subtype-specific biomarker. The method has strong reusability and can also be used to identify other cancer subtypes. In addition, HSSG can be used for other applications besides cancer subtype identification. For example, as HSSG can effectively mine biomarkers that identify cancer subtypes, this means that HSSG can also be applied to the mining of various cancer biomarkers (treating normal samples as subtypes and using HSSG to mine biomarkers in cancer samples versus normal samples), which will be important for cancer-related research. Cancer biomarkers identified by HSSG can also be used as a guide for cancer screening techniques, such as liquid biopsy-based cancer screening.

Although HSSG has the above advantages, it also has some shortcomings: on the one hand, the number of key genes in constructing a gene–gene network is judged subjectively according to the accuracy curve, which may not show the optimal effect of the model; On the other hand, the proposed method will calculate the heterogeneity score of each gene, which leads to a little high time complexity of the whole model. At the same time, parts of the approach can be studied further: we will find that the methods can find genes that effectively distinguish between different cancer subtypes, but these genes are not necessarily disease-causing. The biological significance of these genes can be further studied. At the same time, the differences between enrichment analysis of HSSG mined genes and enrichment analysis based on differential expression methods are also areas where future research could be focused. In future work, we will combine the biological regulation process, the pathogenic genes of cancer, and the genes found by our method to analyze the regulation process in the human body.

Finally, we found that the identification of cancer subtypes based on the random selection of genes was more accurate than the expected random dichotomization or trichotomization, especially in the “Result of identifying multiple cancer subtypes” section. Here, the phenomenon is discussed and analyzed. On the one hand, clustering samples by randomly selected genes is different from directly classifying the sample at random. Although there are fewer genes that contribute significantly to the classification of cancer subtypes (i.e., fewer genes that are directly associated with cancer subtypes and can be used as biomarkers for cancer subtypes), there are many genes that differ in trace amounts across cancer subtypes (i.e., genes that contribute somewhat less to the identification of cancer subtypes). Due to the complexity of cancer, abnormal expression of some genes may lead to partial expression differences in some other genes. This potentially makes it possible that randomly selected feature clustering does not match the expected outcome of randomly dividing the sample. On the other hand, to explore the distribution of tumor sample data for different cancers, we have carried out a PCA analysis of three cancerous tumors and have drawn a PCA plot for the three cancers, as shown in Figure 12. From the plot, we can see that there are indeed some differences between the three tumor samples. This may be the reason for the high accuracy of randomly selected features in identifying cancer subtypes.

## 5. Conclusions

We proposed a new single gene-based feature selection method for cancer subtype identification and applied it to five cancer samples from TCGA. Calculation of the contribution of a single gene to cancer subtype identification has a significant effect on feature selection. Subsequently, a gene–gene network was constructed to explain the biological significance of the genes further and provide a basis for better mining of genes for cancer subtype identification. In conclusion, a single-gene-based cancer subtype identification offers new prospects for related research.

## Figures and Tables

**Figure 1 cells-11-02456-f001:**
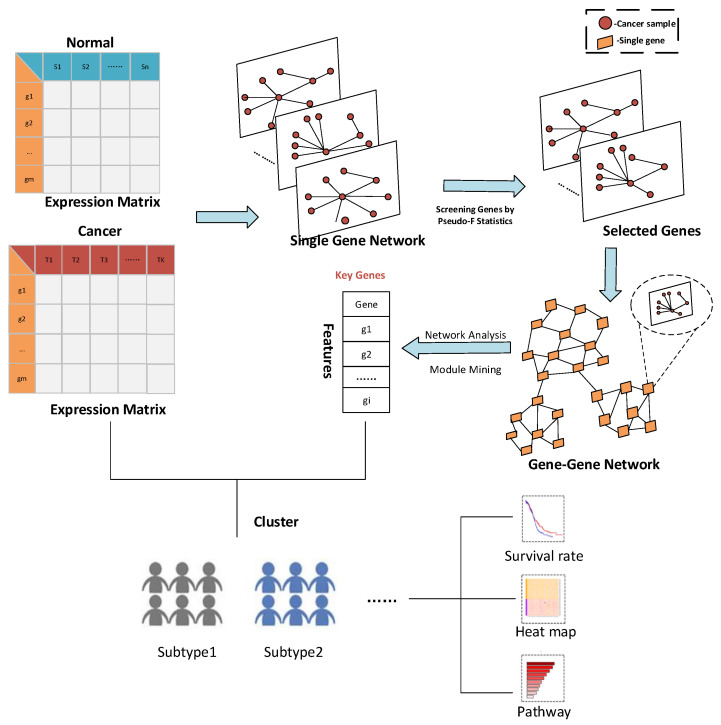
The overall process framework of the HSSG.

**Figure 2 cells-11-02456-f002:**
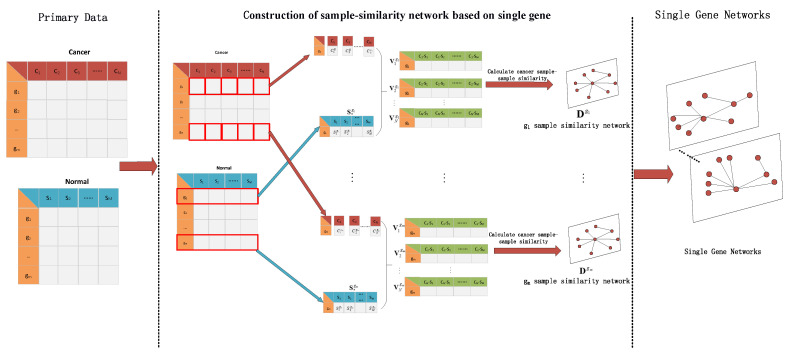
The process for constructing a similarity network of a single gene sample.

**Figure 3 cells-11-02456-f003:**
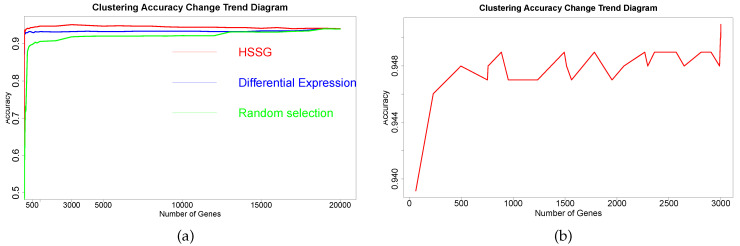
The change curve of cluster accuracy in selecting genes for identifying two subtypes. (**a**) Curve of the accuracy of three different gene screening methods changing with that number of added genes; (**b**) With the addition of gene modules, the accuracy changed with the number of adding genes.

**Figure 4 cells-11-02456-f004:**
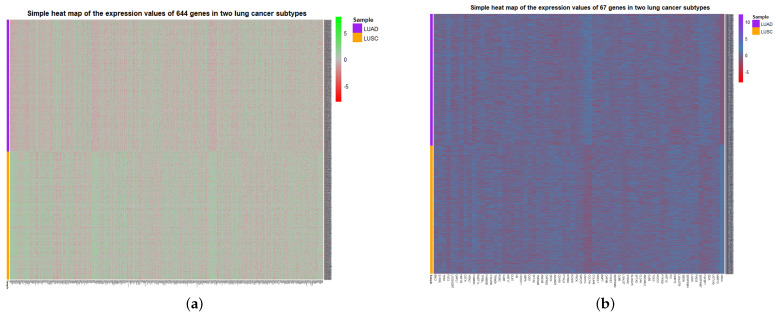
Simple heat map of gene expression for the genes selected in identifying two cancer subtypes. (**a**) The simple heat map of the expression of 644 genes. (**b**) Simple heat map of the expression of 67 genes.

**Figure 5 cells-11-02456-f005:**
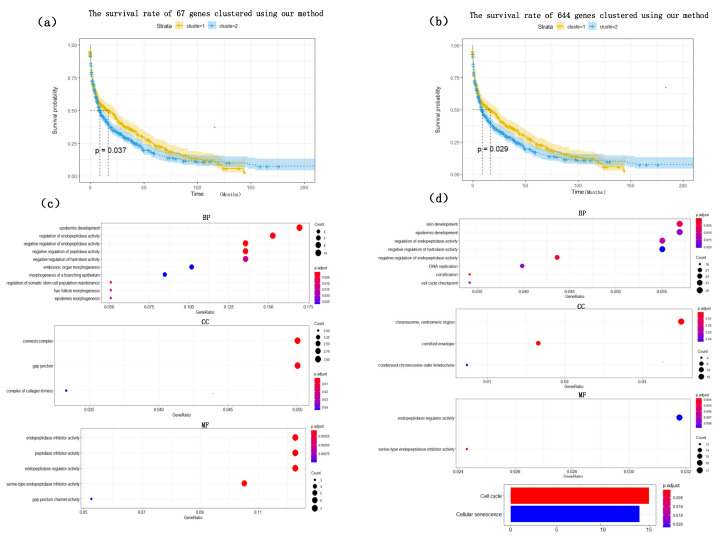
The survival rate and enrichment analysis of 67 genes and 644 selected genes. (**a**) The survival rate schematic of the 67 genes cluster. (**b**) The survival rate schematic of the 644 genes cluster. (**c**) GO enrichment analysis of the 67 genes. (**d**) GO enrichment analysis and KEGG pathway enrichment analysis of the 644 genes.

**Figure 6 cells-11-02456-f006:**
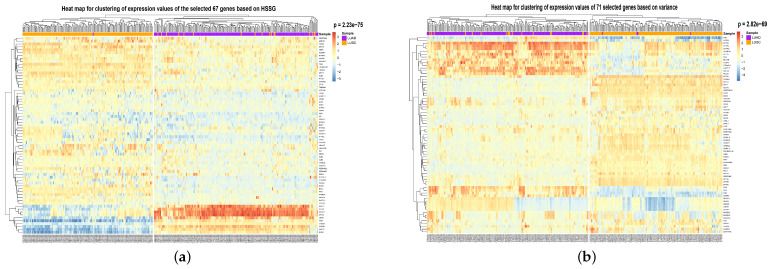
Heat map of gene expression clustering for genes selected by six different methods. (**a**) The heat map of HSSG-selected gene expression clustering. (**b**) The heat map of Variance-selected gene expression clustering. (**c**) The heat map of Entropy-selected gene expression clustering. (**d**) The heat map of Kruskal-test-selected gene expression clustering. (**e**) The heat map of Differential expression-selected gene expression clustering. (**f**) The heat map of Random Forest-selected gene expression clustering.

**Figure 7 cells-11-02456-f007:**
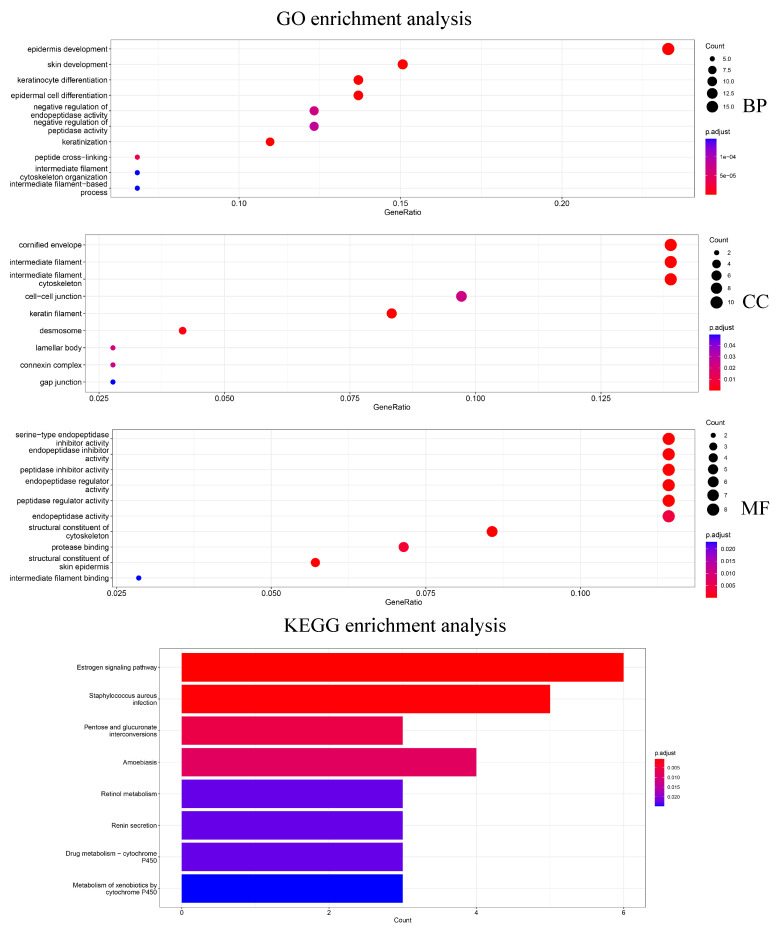
The GO enrichment analysis and KEGG pathway enrichment analysis of differentially expressed screening genes.

**Figure 8 cells-11-02456-f008:**
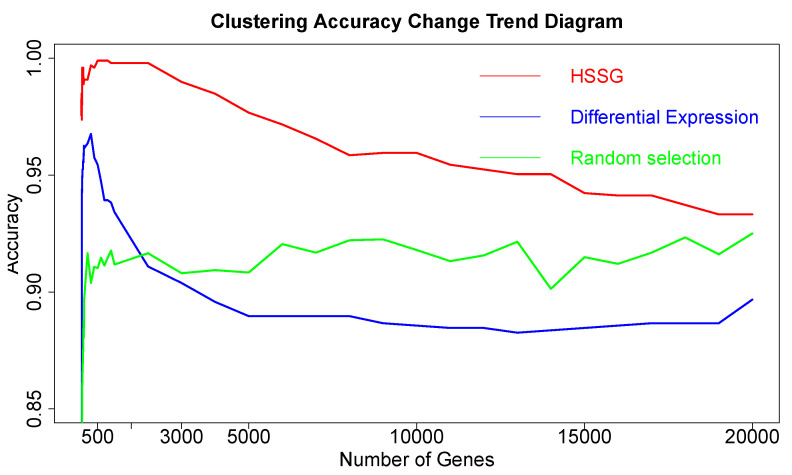
The changing curve of the accuracy of three different methods in identifying three cancer subtypes.

**Figure 9 cells-11-02456-f009:**
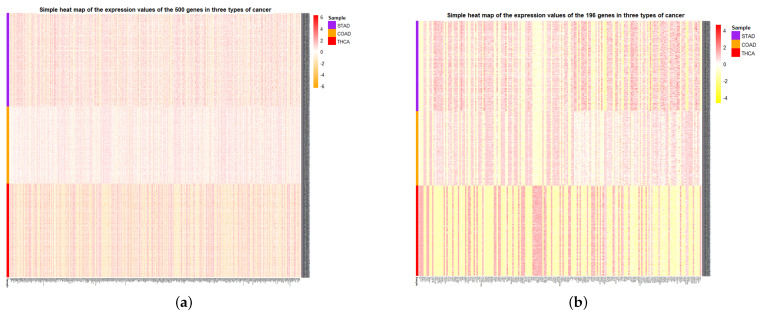
Simple heat map of gene expression for the selected genes in identifying multiple subtypes. (**a**) Simple heat map of 500 genes selected by pseudo-f statistics. (**b**) Simple heat map of the 196 genes selected by network module mining.

**Figure 10 cells-11-02456-f010:**
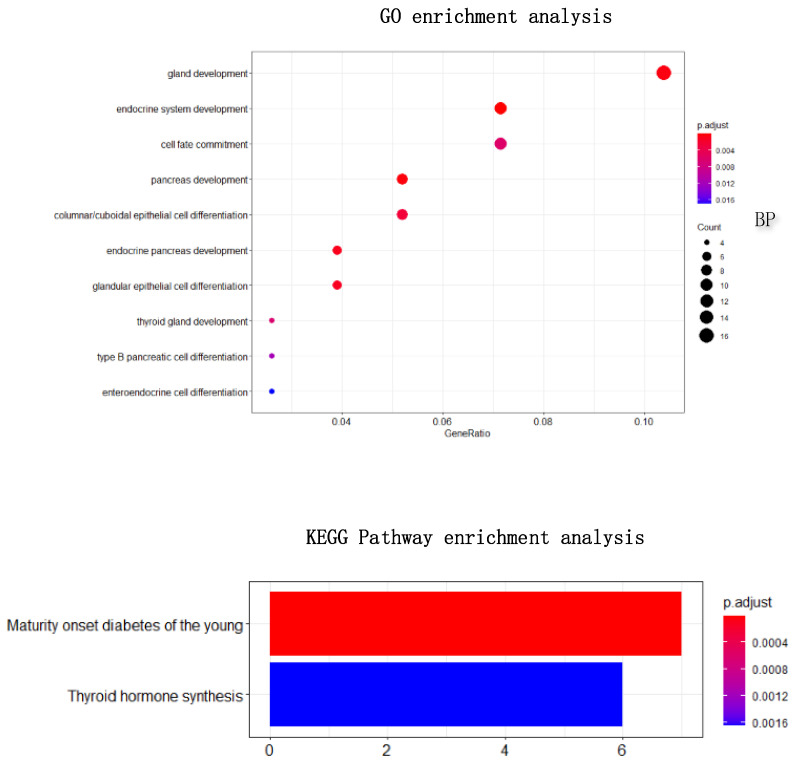
The GO and KEGG pathway enrichment analysis of 196 obtained gene.

**Figure 11 cells-11-02456-f011:**
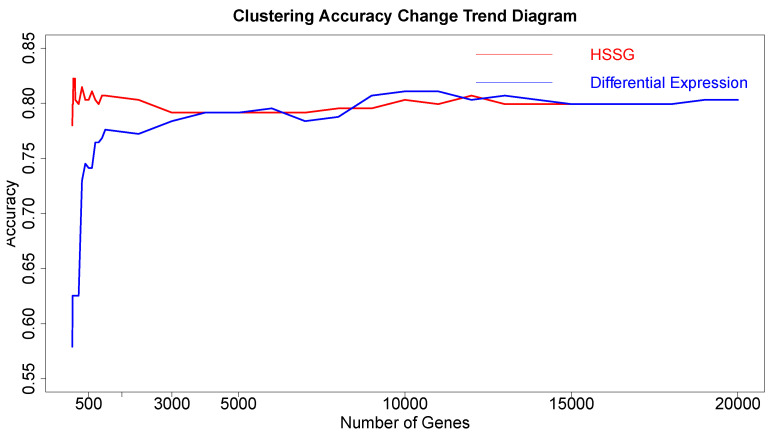
The changing cluster accuracy curve for two different ways to add genes.

**Figure 12 cells-11-02456-f012:**
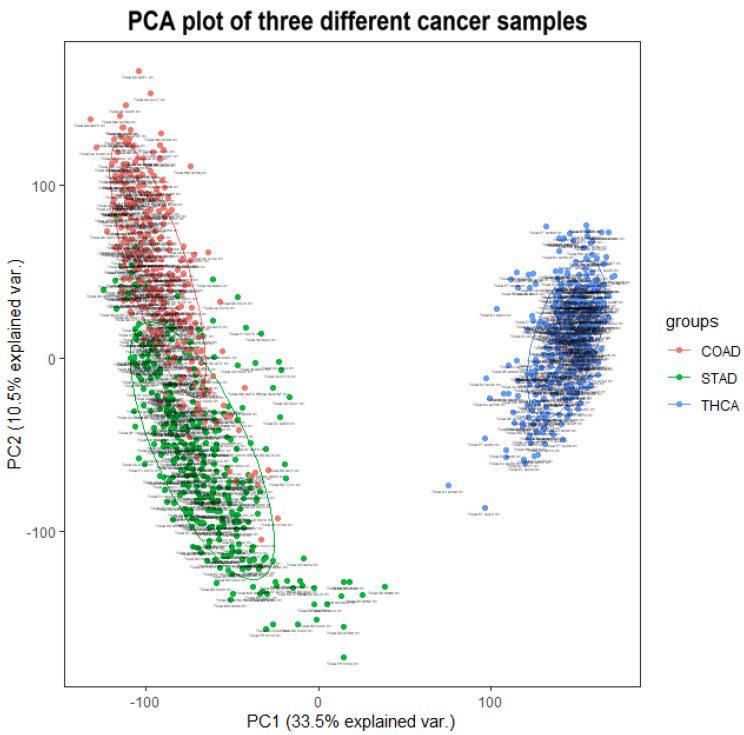
The PCA plot of three different cancer samples.

**Table 1 cells-11-02456-t001:** Number of samples and genes in five types of cancer.

ID	Name	Data Type	Sample		Total Genes
			Tumor	Normal	
LUAD	Lung Adenocarcinoma	RNA-seq	517	56	2050
LUSC	Lung Squamous Cell Carcinoma	RNA-seq	502	54	
STAD	Stomach Cancer	RNA-seq	350	46	
COAD	Colon Cancer	RNA-seq	288	41	
THCA	Thyroid Cance	RNA-seq	513	59	
LUAD	Lung Adenocarcinoma	miRNA-seq	518	46	1881
LUSC	Lung Squamous Cell Carcinoma	miRNA-seq	478	45	

**Table 2 cells-11-02456-t002:** Comparison with six popular feature scoring methods in selecting 67 genes.

Method	Accuracy	Sensitivity	Specificity
Random selection	70.5%	68.6%	73.1%
Differential Expression selection	92.8%	97.5%	89.8%
Variance-based score	93.0%	97.9%	89.1%
Kruskal–Wallis Test	93.6%	98.2%	89.9%
Entropy-based score	93.4%	98.2%	89.6%
Random Forest	93.9%	98.0%	**90.5**%
HSSG	**93.91**%	**98.5**%	90.2%

**Table 3 cells-11-02456-t003:** Comparison with six popular feature scoring methods in selecting 644 genes.

Method	Accuracy	Sensitivity	Specificity
Random selection	90.8%	92.6%	88.6%
Differential Expression selection	93.2%	98.4%	89.1%
Variance-based score	94.3%	97.8%	91.2%
Kruskal-Wallis Test	94.4%	98.2%	91.2%
Entropy-based score	94.3%	98.4%	90.7%
Random Forest	94.5%	98.4%	91.2%
HSSG	**94.7**%	**98.9**%	**91.2**%

**Table 4 cells-11-02456-t004:** Overlap of the six popular methods with the genes selected by the HSSG.

Method	Gene Number	Accuracy	Overlap Number with HSSG	Overlap Rate
Random selection	67	70.5%	0.6 *	0.89%
Differential Expression	79	93.2%	40	50.6%
Variance-based score	71	93.1%	9	12.6%
Entropy-based score	176	93.7%	39	22.1%
Kruskal–Wallis Test	68	93.7%	18	26.0%
Random Forest	81	**93.91**%	24	29.6%
HSSG	67	**93.91**%	67	100%

* The average overlap number of 50 randomly selected.

**Table 5 cells-11-02456-t005:** Comparison with six popular feature scoring methods in selecting 500 and 196 genes.

Method	Gene Number	Accuracy
Random selection	500	81.4%
Differential Expression selection	500	95.4%
Variance-based score	500	96.5%
Kruskal–Wallis Test	500	97.4%
Entropy-based score	500	96.6%
Random Forest	500	99.6%
HSSG	500	**99.8**%
Random selection	196	80.0%
Differential Expression selection	196	96.1%
Variance-based score	196	95.9%
Kruskal–Wallis Test	196	97.1%
Entropy-based score	196	97.8%
Random Forest	196	99.6%
HSSG	196	**99.8**%

**Table 6 cells-11-02456-t006:** Performance of five popular cancer subtypes identification methods.

Method	Accuracy without Feature Selection	Accuracy after Feature Selection	Increase of Method
COCA [35]	66.8%	92.7%	**+25.9**%
LRAcluster [36]	92.8%	94.6%	**+1.8**%
ConsensusClustering [37]	94.1%	94.8%	**+0.7**%
iClusterBayes [38]	94.2%	94.8%	**+0.6**%
IntNMF [39]	93.9%	95.1%	**+1.2**%

**Table 7 cells-11-02456-t007:** The detail of the top 10 miRNA in our study.

Number	Name of miRNA	Heterogeneity Score	Verified
1	hsa-mir-205 [40,41]	1847.826056	**Yes**
2	hsa-mir-149 [42,43]	856.438976	**Yes**
3	hsa-mir-708-5p [44]	684.3413755	**Yes**
4	hsa-mir-203a-3p [45]	542.132398	**Yes**
5	hsa-mir-769-5p [46]	457.3321947	**Yes**
6	hsa-mir-326 [47]	449.3204298	**Yes**
7	hsa-mir-6510	424.1914996	**No**
8	hsa-mir-6512	387.6342542	**No**
9	hsa-mir-378a-3p [48]	375.8590445	**Yes**
10	hsa-mir-1271-5p [49]	356.288151	**Yes**

## Data Availability

All datasets used in this study are accessible from UCSC Xena (https://xenabrowser.net/datapages/, (accessed on 10 June 2022).

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
