# Peer review of "HSSG: Identification of Cancer Subtypes Based on Heterogeneity Score of A Single Gene"

_cells, 2022, doi:10.3390/cells11152456_

Round 1

Reviewer 1 Report

Pang & Wu et al. have developed a method to determine cancer (sub) type based on heterogeneity scores of genes.  Feature selection is a well know variable in biomedical research, both to identify biological mechanisms and to identify prognostic and predictive markers. The authors have performed a series of experiments including survival, clustering, and pathway analyses for HSSG. Although the approach seems very interesting and promising, based on the study design and results it is not clear what the difference and overlap is with the established approaches. I am curious to learn about the HSSG performance for survival, clustering, and pathway analyses compared with established methods.  

Please find below my detailed observations, containing suggestions to improve this research.

Introduction:

-The introduction, especially the formulation of the research gap, and the solution provided by the authors seem not to align. 

-The introduction could be written with more care from a clinical point of view. For example.  “In the cancer treatment process, it has been found that patients with the same clinical manifestations will have different reactions after receiving the same treatment, which indicates that the traditional cancer classification method is based on where the cancer cells are located, a vague definition of cancer categories” This seems a strongly generalized statement based on assumptions. The term clinical manifestations was not defined and here, the authors assume that cancer treatment is based on clinical manifestations (whatever this might be) and that the treatment reaction apparently is used to determine a cancer category? For breast cancer, for example, molecular subtyping is used to provide valuable information for treatment decisions. The introduction requires more explanation, possibly consulting a clinician.

Methods:

-Please, provide the code the authors wrote. Preferably on GitHub to allow other researchers to use this approach. If this is not possible, the code should at least be provided for the review process in order to allow a reviewer to reproduce the main findings.

-How are the tumor types selected for this study? TCGA contains many… It seems like the authors aim to identify cancer types rather than cancer subtypes. For example, would this technique be able to more accurately identify breast cancer subtypes including Luminal, Basal, and ERBB2?

-Please, insert an overview of R packages used including version numbers.

-What is meant by ”similarity of any two cancer samples under gene g is calculated”

-How are the training and validation sets determined?

Results:

-Please, include a table summarizing the different approaches including differential expression and other commonly used approaches together with HSSG. Provide the number of features selected, the overlap with the other approaches, the performance/survival etc. to allow the reader to compare these

-The accuracy of random selection (figure 3) is very high (70.5 %) as one would expect an accuracy of 50%. It seems like the data used is not balanced. Please define the dataset in more detail.

-Also, please extend the y-axes in a way that x includes 0 (figure 3a). It is not clear what is the difference between a and b.

-It is not clear to me why are 67 and 644 genes were selected (table 2 and 3)? It would probably be fairer to compare the different methods by AUC instead of accuracy since accuracy is the cutoff depended.

-The authors aim to identify biological significance by enrichment of the selected genes. To what extent does the enriched set overlap or differ from the enriched set derived by the commonly established methods such as by differential expression or LASSO?

-The heatmaps are not readable. Title and axis labels are missing and no clustering was performed. The names on the x-axes seem not similar to the number of features. What are these names?  What is the point the authors aim to make by showing these heatmaps? At this point, they seem non-informative.

-The authors conclude that the KEGG pathway analysis result is consistent with the general signaling pathway of cancer samples. This requires more explanation. For example, how is this different from other approaches or compared with the random selection? Based on this experiment it is impossible to determine the biological relevance of this new feature selection approach.

-Table 4: How can the clustering accuracy of random selection be this high?

-Figure 7: Legend, title, and axes titles are missing. Clustering and comparative analyses (differential expression) would be helpful to understand the results.

-Figure 8: “Since the purpose of this experiment was to effectively distinguish gastric cancer, colon cancer, and thyroid cancer, enrichment analysis  could show the correctness of genes selected by HSSG, on the other hand.” How does figure 8 support this conclusion? It is unclear how this is related and why this is specific for these cancer types.

-Table 5: are these the optimal accuracies of aiming at a certain sensitivity or specificity?  included?

 Discussion & conclusion:

This manuscript focuses on cancer type identification. It would be valuable if the authors discuss the different situations in which this new technique could be beneficial and when it would make more sense to rely on other approaches. Also, purposes could benefit from this technique besides tumor type identification? For example, liquid biopsy-based cancer screening? Etc.

Some minor general points:

- In general, sentences are long and incorrect grammar was used resulting in sentences that are not understandable. For example “The heat map that there was a significant difference in the expression of genes screened by HSSG  in the samples of two lung cancer subtypes, which is consistent with the expected results.”

-Is reference 3 correct? “For example, in 2012, Chen et al. used a comprehensive analysis of multi-omics data such as copy number 25 and gene expression to discover the molecular drivers of breast cancer and effectively provide a new breast cancer subtype stratification[3].”  Ref 8 is referring to Chen et al, ref 3 is Curtis et al.

-“…they use the known biological network without considering to use of the 67 network constructed by the heterogeneous genes.” This requires more explanation

-“the network constructed with high-heterogeneous genes maybe 69 more significant in mining cancer subtype-specific biomarkers.” Why is this the case?

-The figures require more explanation including titles, axes-labels, and descriptions

-Row 250: what means “clustering25”

-I suggest the use of the word ordering or ranking instead of sequencing since sequencing often refers to determining the DNA of RNA sequence instead.

Reviewer 2 Report

In this manuscript, the authors proposed a method for feature selection to identify cancer subtypes based on the heterogeneity score of a single gene. The study is interesting, however, a number of issues should be addressed.

Major issues:
1.The definitions of symbols in Section 2.2 are inconsistent or undefined, for example, the symbols of "the gene expression value of j-th normal sample under the gene g" and "the gene expression value of i-th cancer sample under the gene g" were inconsistent, the s and t were not defined in the formula 2.

2. The symbols in Figure 2 should be marked with subscripts, and the preceding symbols C_ig, S_jg, V_ig, S_*g, D_g should be added to Figure 2.

Minor issues:
1. There are errors in citing the literature. For example, reference 3 on lines 25-27 is not the work of Chen et al. (2012).

2. There are typos in draft. For example, line 237.

3. p.3, is 'Lung Adenocarcinoma" abbreviated as LADC or LUAD?

Round 2

Reviewer 1 Report

The authors have improved both the readability and the research quality significantly. I appreciate the clarification in both the manuscript and cover letter. The introduction is now in line with the research presented. Also, it serves the scientific community that the authors have uploaded the source code on GitHub and will likely have a positive impact on the use of this new technique by others. Most of the raised points have been addressed properly. However, I have some (major and minor) concerns that are not fully addressed yet. 

Major:

- As pointed out in the previous review, another intuitive and visual way of comparing the genes selected for the six different methods is by comparing the unsupervised clustering of the heatmaps for the different methods. For example, use the panels as presented in table 4 by the different methods and visualize them in heatmaps including unsupervised clustering for both the samples and genes and quantify the separation by Fisher's p-value. According to previously described results, HSSG should perform most optimally.

-Please be consistent in the five control/compare methods used throughout the manuscript. For example, “Random selection” is presented in table 2 and 3 but missing in table 4. Table 5 shows the results of random selection and differential expression but the other three methods are not presented.

-Not all adjustments as described in the cover letter are included in the manuscript text. For example “Figure 3a shows the clustering accuracy curve with the number of genes added, based on the ranking of each gene's contribution (pseudo-F statistic value), with the constant addition of genes.” Please, make sure that all the proposed changes are correctly presented in the manuscript.

-It is still not completely clear how the training and validation sets are determined. Has this been random? Did the authors use the sample() function in R and used set.seed() in order to make random selection reproducible?

-“How can the clustering accuracy of random selection be this high?” This point needs to be addressed properly before this study can be published.

The authors indicate that the difference may be due to inherent differences. I do not know the data, but the only reasonable explanation I can think of is that there is a confounding bias, such as another protocol used or processed at different centers. My suggestion is two-fold: First, please sort out if there are any indications that the samples from the different tumor types were treated differently, both from a technical perspective such as laboratory protocol or from a medical perspective such as tumor stage of treatment conditions. A PCA plot might give some valuable information. Second,  this point should be described in the discussion section.

Minor:

-Please, remove the sentences discussing the results from GO and KEGG analysis from the results section and discuss these in the discussion section. For example row 327.

- “In addition, 79 genes selected by the HSSG “, row 318. Should this be selected by Differential Expression?

-Please, have a careful look and check the entire manuscript text and remove redundant text such as lines 133 and 434, missing spaces, for example in the description of figure 4, etc.

-Code:

*some parts of the code are missing in order to redo the analysis:

      Set the working directory

-       Create the folders “file” and “similar”

-       *NGennet<-NGennet[1:500,1:500] generates an error (Error in `vectbl_as_col_location()`)

-       *I suggest using set.seed() to make random selection reproducible

-       *There are some misspellings in the code such as “damo” instead of “demo”

-       *Some more comments in the code would be helpful to understand the different processing steps

-       *Based on the code, not all R packages are listed in the manuscript text, and also the R version used is not mentioned. Please add this information. 
